# A Novel Gammapartitivirus That Causes Changes in Fungal Development and Multi-Stress Tolerance to Important Medicinal Fungus *Cordyceps chanhua*

**DOI:** 10.3390/jof8121309

**Published:** 2022-12-16

**Authors:** Qiuyan Zhu, Najie Shi, Ping Wang, Yuxiang Zhang, Fan Peng, Guogen Yang, Bo Huang

**Affiliations:** 1Anhui Provincial Key Laboratory of Microbial Pest Control, Anhui Agricultural University, Hefei 230036, China; 2School of Plant Protection, Anhui Agricultural University, Hefei 230036, China

**Keywords:** *Cordyceps chanhua*, gammapartitivirus, fruiting body, horizontal transmission, stress tolerance

## Abstract

Cicada flower, scientifically named *Cordyceps chanhua*, is an important and well-known Chinese cordycipitoid medicinal mushroom. Although most mycoviruses seem to induce latent infections, some mycoviruses cause host effects. However, the effects of mycovirus on the fungal development and stress tolerance of *C. chanhua* remain unknown. In this study, we report a novel mycovirus designated Cordyceps chanhua partitivirus 1 (CchPV1) from *C. chanhua* isolate RCEF5997. The CchPV1 genome comprises dsRNA 1 and dsRNA 2, 1784 and 1563 bp in length, respectively. Phylogenetic analysis using the aa sequences of RdRp revealed that CchPV1 grouped with members of the genus *Gammapartitivirus* in the family *Partitiviridae*. We further co-cultivated on PDA donor strain RCEF5997 and recipient *C. chanhua* strain RCEF5833 (Vf) for 7 days, and we successfully obtained an isogenic line of strain RCEF5833 with CchPV1 (Vi) through single-spore isolation, along with ISSR marker and dsRNA extraction. The biological comparison revealed that CchPV1 infection slows the growth rate of the host, but increases the conidiation and formation of fruiting bodies of the host. Furthermore, the assessment of fungal tolerance demonstrated that CchPV1 weakens the multi-stress tolerance of the host. Thus, CchPV1 infection cause changes in fungal development and multi-stress tolerance of the host *C. chanhua*. The findings of this study elucidate the effects of gammapartitivirus on host entomogenous fungi and provide a novel strategy for producing high-quality fruiting bodies of *C. chanhua.*

## 1. Introduction

Cicada flower, the asexual fruiting body formed from infected cicada nymphs, is an important and well-known Chinese cordycipitoid medicinal mushroom with a history of use for about 1500 years [1]. The mushroom has long been regarded by the Latin name *Isaria cicadae*. However, with the recent discovery of the teleomorph of this species, as well as morphological and multigene phylogenetical analyses, *Cordyceps chanhua* in the family *Cordycipitaceae* was proposed and accepted as the scientific name of Cicada flower, derived from the ancient Chinese name “chanhua” [2]. For long time, *C. chanhua* has been used as a food, tonic, and folk medicine to treat malaria, palpitations, cancer, fever, diabetes, eye diseases, dizziness, and chronic kidney diseases [3]. Furthermore, the mushroom contains bioactive compounds similar to those found on *C. sinensis* and *C. militaris*, which are the most studied and widely used in the world [3]. To meet the great consumption demand of cicada flowers, large-scale cultivation of the coremium of *C. chanhua* has been widely explored and successfully developed in China [4]. A series of commercial products derived from chanhua have been developed and used as functional foods in China [5]. Commercial products from *C. chanhua* are also sold as health supplements, such as “Cikaria” in Sweden [3].

Mycoviruses are widely distributed in almost all fungi, including entomopathogenic fungi. Most known species of mycoviruses have double-stranded (ds) RNA genomes. The International Committee on Taxonomy of Viruses (ICTV) (https://ictv.global/taxonomy, accessed on 14 November 2022) classifies dsRNA mycoviruses into one genus and eight families, including *Partitiviridae*. Furthermore, partitiviruses with bisegmented dsRNA genomes of 3–4.8 kb are generally encapsidated in isometric particles about 25–40 nm in diameter, and partitivirus capsids were shown to have a so-called “*T* = 2” organization comprising 60 CP dimers arranged on a *T* = 1 icosahedral lattice, as determined by cryo-electron microscopy and three-dimensional (3D) image reconstruction, as well as X-ray crystallography [6,7,8,9]. The *Partitiviridae* includes plant, fungal, and protozoa viruses and has five recognized genera, namely, *Alphapartitivirus*, *Betapartitivirus*, *Deltapartitivirus*, *Gammapartitivirus*, and *Cryspovirus* [10], as well as two proposed genera, “*Epsilonpartitivirus*” and “*Zetapartitivirus*” [11]. All known members of the genus *Gammapartitivirus* infect ascomycetous fungi. To date, at least 61 species of *Gammapartitivirus* have been identified.

Although the vast majority of mycoviruses seem to induce latent infections, some mycoviruses cause host effects [12]. Moreover, some mycoviruses may have a positive impact on their hosts, such as the enhancement of virulence in plant pathogenic fungi [13] and an increase in competitive ability in some yeasts [14], while some mycoviruses cause unfavorable phenotype alterations, such as reduced growth, low conidiation, and hypovirulence. Partitiviruses called cryptic viruses are classically considered to have few, if any, deleterious effects on host cells [12]. However, several recent examples showed that partitivirus interaction with fungal phytopathogens are more likely to be hypovirulence agents [15]. Most gammapartitiviruses are associated with latent infections, but Aspergillus fumigatus partitivirus 1, a related unclassified virus, influences the biological traits of the host, such as aberrant phenotypic alterations and attenuation of growth of the fungus. For insect pathogenic fungi, early work on *Metarhizium anisopliae* revealed that mycovirus mediated hypovirulence in host with a decrease in endochitinase secretion and conidiospore production, while recent studies on *Beauveria bassiana* confirmed that mycovirus infections enhanced the fitness of host by increasing conidiation, stress tolerance, and virulence (Shi et al., unpublished). However, the effects of partitiviruses on fungal development and stress tolerance of *C. chanhua* remain unknown.

Previously, we found that, in two *C. chanhua* isolates, RCEF5833 and RCEF5997, there exist significant differences in growth rate and conidiation capacity. Furthermore, detection of dsRNA mycovirus showed that isolate RCEF5997 harbors three dsRNA fragments, while no dsRNA mycovirus was found in isolate RCEF5833. This phenomenon indicates that mycovirus infection could affect the biological traits of *C. chanhua*. In this study, we characterized a novel mycovirus, namely, Cordyceps chanhua partitivirus 1 (CchPV1) from isolateRCEF5997, and analyzed its phylogenetic relationships with members of the *Partitiviridae* family. We also determine the effects of viral infection on *C. chanhua* in fungal development and stress tolerance.

## 2. Materials and Methods

### 2.1. Fungal Isolates and Culture

The *C. chanhua* virus donor strain RCEF5997 and recipient strain RCEF5833 were isolated from bamboo cicadas *Platylomia pieli* Kato (Hemiptera: Cicadidae) in Anhui Province, China. *C. chanhua* strains were cultured on potato dextrose agar (PDA, 20% potato, 2% glucose, and 2% agar, *w*/*v*) in Petri dishes (6 cm diameter) for 10 days, and stored in 20% (*v*/*v*) glycerol at −80 °C. The isolates were grown on PDA plates overlaid with a sterilized cellophane membrane to collect fresh mycelium. Conidial suspensions were obtained by vortex-mixing in 0.05% (*v*/*v*) Tween-80 aqueous solution; the spore concentration was determined using hemocytometer.

### 2.2. dsRNA Extraction and Purification

About 500 mg of fresh mycelium of *C. chanhua* isolates was collected and ground in liquid nitrogen. dsRNAs were extracted using CF-11 cellulose (Sigma) chromatography [16,17]. To eliminate contaminating DNA and ssRNA, the dsRNA samples were treated with DNase I and S1 nuclease (TaKaRa, Dalian, China). Purified dsRNA was detected by agarose gel (1.5%, *w*/*v*); electrophoresis was run in TAE buffer (0.04 M Tris-base, 0.02 M CH_3_COOH, 0.001 M Na_2_EDTA, pH 8.0) and visualized after staining with ethidium bromide.

### 2.3. RNA-Sequencing, Molecular Cloning, and Sequence Analysis

dsRNAs of *C. chanhua* strain RCEF5997 were sequenced on an Illumina HiSeq 2500 platform at BGI (Shenzhen, China). A de novo assembly of reads was carried out using Trinity (v2.1.1) [18]. Consequently, contigs were achieved. All contigs were subjected to BLASTx analysis (https://blast.ncbi.nlm.nih.gov/Blast.cgi, accessed on 14 November 2022).

The terminal sequences of CchPV1 were determined using the RNA ligase-mediated rapid amplification of cDNA ends (RLM-RACE) [19]. All cDNA amplicons were cloned into the pMD18-T vector (TaKaRa, Dalian, China) and introduced into *E. coli* TOP10 for sequencing at least three times. All the primers used in this study are listed in Appendix A.

Putative ORFs of CchPV1 were predicted using ORFfinder (https://www.ncbi.nlm.nih.gov/orffinder, accessed on 14 November 2022). Multiple sequence alignment of amino-acid sequences was performed using MAFFT [20]. Phylogenetic analysis of CchPV1 was computed by MEGA-X using the maximum likelihood (ML) method with 1000 replicates, and the LG + G + I + F substitution model was used [21].

### 2.4. Horizontal Transmission of CchPV1

To obtain isogenic CchPV1-infected and CchPV1-free strains, *C. chanhua* strain RCEF5997 (CchPV1-infected; donor) and *C. chanhua* strain RCEF5833 (CchPV1-free; recipient) were co-cultivated on PDA at 25 °C for 7 days. Following contact, the mycelial agar plugs from the margin of the recipient were re-cultured onto fresh PDA medium until sporulation; subsequently, isolation of single spores was carried out. To distinguish the donor and recipient isolates, a total of 12 inter-simple sequence repeat (ISSR) primers applicable to the phylogenetic relationships among *C. chanhua* strains were screened [22]. Horizontal transmission of CchPV1 to RCEF5833 was confirmed by dsRNA extraction and RT-PCR amplification using special virus primers.

### 2.5. Phenotype Assays

For growth assays, conidia from strains RCEF5833 without CchPV1 (Vf) and RCEF5833 with CchPV1(Vi) (1 μL from 1 × 10^7^ conidia/mL suspensions) were inoculated at the center of PDA, SDAY (1% *w*/*v* peptone, 4% *w*/*v* dextrose, 0.2% *w*/*v* yeast, 1.5% *w*/*v* agar), and 1/4 SDAY plates. At 10 days post inoculation (dpi), all colony diameters were measured as indices of radial growth rates (colony diameter), using the cross-crossing method [23]. All experiments were performed in triplicate.

To assay the conidiation capacity of the Vf and Vi strains, 70 μL of a suspension of 10^6^ conidia/mL was evenly spread on PDA plates (6 cm diameter) and cultured in the light at 25 °C for 10 days. The conidia were washed off into 30 mL of 0.05% Tween-80 to release conidia by vortex-mixing under 10 min of supersonic vibration. The concentration of the conidial suspension was determined by counting the conidia using a hemocytometer under a microscope [24]. All experiments were performed in triplicate.

For the conidial germination assay, 20 μL of conidial suspension (5 × 10^6^ conidia/mL) was spread on PDA medium, followed by incubation at 25 °C for 24 h. The conidial germination was observed by microscope (Olympus BX 51, Tokyo, Japan); from 4 h onward, the percentage of germinated conidia on each plate was assessed every 4 h by counting the number of germinated conidia that could be seen under the microscope (always >300 conidia per treatment) [25]. All experiments were performed in triplicate.

To assay the formation of fruiting bodies, a 1 mL conidial suspension (1 × 10^5^ conidia/mL) of Vi or Vf strains was injected into the hemocoel of Chinese tussah silkworm pupae. To form fruiting bodies, incubation was performed at 17 °C in the dark for 14 days, at 20 °C in the light for 12 h, at 16 °C in the dark for 12 h for 5 days, at 25 °C in the light for 16 h, and in the dark for 8 h for about 14 days. The assay was performed in triplicate, with 20 insects per replicate for each condition [26].

For chemical stress tolerance assays, conidial suspensions (1 µL; 1 × 10^7^ conidia/mL) of Vf and Vi were cultured on PDA plates with supplementary chemical reagents, including the cell-wall-disturbing compound Congo red (1 mg/mL), H_2_O_2_ (10 mM) as an inducer of oxidative stress, and NaCl (0.5 M) as an inducer of osmotic stress, and then incubated in the light for 10 days at 25 °C. Colony diameter was measured as described earlier, and PDA plates (without any modifications) were used as the control [27]. All experiments were performed in triplicate.

For UV irradiation assays, the conidial tolerance of each mutant was assayed using a previously described method [28]. Briefly, these plates with conidial suspensions (10 µL, 5 × 10^6^ conidia/mL) of Vf and Vi were exposed to an irradiance of 312 nm wavelength at 100 µJ·cm*^−^*^2^ using an HL-2000 Hybrilinker (UVP, Upland, CA, USA). After exposure, germination was observed using an Olympus BX 51 (Tokyo, Japan) at 24 h. The relative percentage germination was assessed using the previously mentioned methods. All experiments were performed in triplicate.

For thermotolerance assays, conidial suspensions (5 × 10^6^ conidia/mL) of Vf and Vi strains were incubated in a water bath at 40 °C for 1 h, and then 10 µL of the suspension was placed on PDA plates. Germination was observed using an Olympus BX 51 (Tokyo, Japan) at 24 h, and the relative germination rates were calculated to allow comparisons of the number of germinated conidia with and without heat stress. All experiments were performed in triplicate.

For the penetration assay, a 1 µL conidial suspension diluted by 0.05% Tween-80 (1 × 10^7^ conidia/mL) was added onto the surface of a cicada wing placed on a PDA plate. After incubation at 25 °C for 3 days, the cicada wing was removed, and the PDA plate was further incubated to allow the growth of the fungi that had penetrated the insect wing and reached the nutrient-rich PDA medium. If the fungi successfully penetrated the intact insect cuticle, a colony would appear on the PDA plate 3–5 days after removal of the locust wing. All experiments were performed in triplicate.

### 2.6. Statistical Analysis

All data were analyzed using GraphPad Prism version 7.0 and SPSS v23.0 software (SPSS Inc., Chicago, IL, USA). Subsequently, data (mean ± SE) from different experimental groups were analyzed using Student’s t-test and one- or two-way analysis of variance (ANOVA) followed by a least significant difference (LSD) test. A *p*-value of 0.05 or lower was considered statistically significant.

## 3. Results

### 3.1. dsRNA Segments in C. chanhua RCEF5997

During studies on the biological properties of *C. chanhua* isolates, a remarkable phenotypic plasticity was observed between isolates RCEF5833 and 5997 incubated under the same conditions, and the conidial production of RCEF5997 was more than twofold that of RCEF5833, while the growth rate of RCEF5997 was reduced by 5.8% (*p <* 0.05) compared to the RCEF5833 strain. We further extracted the dsRNA of these two strains and found that isolate RCEF5997 harbors three dsRNA fragments (dsRNA 1, dsRNA 2, and dsRNA 3), while no dsRNA was detected in isolate RCEF5833 (Figure 1A–C). Following nucleotide sequence analysis from RNA-seq data, we identified two partitivirus-like contigs (Figure 1D). Unfortunately, we were not able to obtain any sequence information on dsRNA 3. The genome sequence of the mycovirus was further determined by RT-PCR, and RACE cloning. The mycovirus from isolate RCEF5997 was named “Cordyceps chanhua partitivirus 1” (CchPV1). The genome sequence of CchPV1 was deposited in the GenBank database (accession numbers OP727721 (dsRNA 1) and OP727722 (dsRNA 2)).

### 3.2. Genome Structure of CchPV1

The CchPV1 genome comprises two segments (dsRNA 1 and dsRNA 2) (Figure 2A). The dsRNA 1 and dsRNA 2 genome segments of CcPV were found to be 1784 and 1563 bp in length with a G + C content of 51.3% and 54.3%, respectively. Each segment had a single open reading frame (ORF). The ORF of CchPV1 dsRNA 1 (nt 69–1691) encodes a putative 540 aa (62.8 kDa) long RNA-dependent RNA polymerase (RdRp), while the ORF of CchPV1 dsRNA 2 (nt 113–1420) encodes a putative 435 aa (46.9 kDa) coat protein (CP). The 5′-untranslated region (5′-UTR) and 3′-UTR of CchPV1 dsRNA 1 are 68 and 93 bp long, respectively, while those of CchPV1 dsRNA 2 are 112 and 143 bp in length, respectively (Figure 2B). The 5’-UTRs of CchPV1 dsRNA 1 and dsRNA 2 are highly conserved, with the conserved element (GCCUCCUUCAUC). Similarly, the 3’-UTRs of the two CchPV1 genome segments share common conserved regions (UGUGNGAANC). However, the distinct poly(A) was not detected in CchPV1 dsRNA 1 or dsRNA 2 (Figure 2C).

### 3.3. Taxonomic and Phylogenetic Position of CchPV1

A homology search of the GenBank database using BLASTp showed that CchPV1 CP revealed the highest identity (54.15%) with that of Penicillium digitatum partitivirus 1. According to BLASTp analysis, the RdRp of CchPV1 had 56.9–70.7% aa sequence identity to the members of the genus *Gammapartitivirus*, with the highest identity of 70.7% to Penicillium stoloniferum virus S (YP_052856.2). Moreover, the conserved domain database (CDD; NCBI) search indicated that an RdRp domain with six conserved motifs typical of members of the family *Partitiviridae* (motifs III to VIII; Appendix A) was found in the RdRP of CchPV1.

To determine the phylogenetic relationship of CchPV1 with other partitiviruses, an ML phylogenetic tree was constructed on the basis of RdRp amino-acid sequences using Beauveria bassiana victorivirus 1 as an outgroup. The phylogeny inferred from the alignment showed seven different clades with high bootstrap support representing five recognized genera, namely, *Alphapartitivirus*, *Betapartitivirus*, *Deltapartitivirus*, *Gammapartitivirus*, and *Cryspovirus*, and two proposed genera, “*Epsilonpartitivirus*” and “*Zetapartitivirus*”. Moreover, the ML phylogenetic tree revealed that CchPV1 grouped with members of the genus *Gammapartitivirus* and sorted with the other genera of the family *Partitiviridae*.

The current cutoff value for species demarcation in the family *Partitiviridae* exhibits less than 90% and 80% aa sequence identity for the RdRp and CP, respectively [10]. Hence, we propose that CchPV1 is a new member of the genus *Gammapartitivirus*. This is the first report on a new partitivirus hosted by the important medicinal fungus *C. chanhua* (Figure 3).

### 3.4. Horizontal Transmission of CchPV1 from C. chanhua RCEF5997

To genetically distinguish between *C. chanhua* strains RCEF5833 and RCEF5997 prior and subsequent to horizontal transmission experiments, inter-simple sequence repeat (ISSR) markers were developed. A total of 12 ISSR primers were screened, and one (P6) of these primers which gave sufficient amplification was selected to discriminate these two strains for the study (Figure 4A, Appendix A).

To obtain isogenic strains of *Cordyceps chanhua* strain RCEF5833 with mycovirus infection, we co-cultivated on PDA donor strain RCEF5997 and recipient strain RCEF5833 (Vf) for 7 days (Figure 4B). A total of 70 single-spore isolates from the intersection of co-cultivated colony were randomly subjected to DNA extraction and ISSR analysis, revealing that 61 single-spore isolates were identified as RCEF5833 (Figure 4C). Furthermore, dsRNAs were extracted from these 61 single-spore isolates, revealing that only one single-spore isolates contained the same dsRNA elements as RCEF5997 (Figure 4D). The presence of CchPV1 in the infected strain RCEF5833 of *Cordyceps chanhua* (Vi) was further confirmed by RT-PCR.

### 3.5. CchPV1 Infection Slows Down the Growth Rate of the Host, but Increases the Conidiation and Formation of Fruit Bodies of the Host

The effect on vegetative growth due to CchPV1 infection was investigated. The growth rate of the Vi strain was reduced by 9.2% (*p <* 0.05), 7.1% (*p <* 0.05), and 20.0% (*p <* 0.05) on SDAY, PDA, and 1/4 SDAY as compared to the Vf strain, respectively. The results indicated that CchPV1 slowed the growth rate of the host (Figure 5A,C).

We next explored the effect of CchPV1 infection on conidial yield of the host. The conidial yields from the 10 day old cultures of the Vi and Vf strains were assessed as 3.8 × 10^7^ conidia /cm^2^ and 2.0 × 10^7^ conidia/cm^2^, respectively, indicating a marked 91% increase in the CchPV1 infection (Figure 5C).

Particularly, the fruiting bodies of *C. cicadae* as a main ingredient have long been utilized in both medicine and food in China. Thus, we evaluated the effect of CchPV1 on asexual fruiting bodies of the host. The mature fruiting bodies were produced after about 33 days of culture on Chinese tussah silkworm pupae. During the entire cultivation period, we observed that formation of the fruiting bodies of the Vi strain grew earlier and more densely than those of the Vf strain (Figure 5B). We further found that the production of the fruiting bodies of 1 g of pupae produced by Vi and Vf was 60.2 mg and 36.7 mg, respectively (Figure 5D). Thus, the Vi strain displayed a significant increase in formation of fruiting bodies by 64% compared with Vf strain.

These results suggest that CchPV1 infection has a more significant effect on asexual reproduction than on vegetative growth of *C. chanhua*.

### 3.6. CchPV1 Infection Weakens Multi-Stress Tolerance of the Host

To examine the effect of CchPV1 on fungal growth under different chemical stress conditions, we assessed the mycelial growth of the Vi and Vf strains on PDA medium containing NaCl, H_2_O_2_, and Congo red (Figure 6A). Compared with the Vf strains, the relative inhibition of Vi growth was decreased by 15.2% (*p <* 0.05) and 35.9% (*p <* 0.01) on PDA containing NaCl and H_2_O_2_, respectively, while the sensitivity to Congo red was reduced by 39.3% (*p <* 0.001) (Figure 6B). These data demonstrate that the osmotic and antioxidant capacity and cell-wall integrity of the Vi strain were significantly reduced compared to the Vf strains.

To evaluate the effect of CchPV1 on the host under UV irradiation and thermal stress, the relative germination rate of conidia exposed to these stresses was investigated 24 h after stress exposure. Compared with the Vf strain, the relative germination rate of the Vi strain was reduced by 27.9% (*p <* 0.001) after UV irradiation for 24 h (Figure. 6C). Similar results were obtained for conidial tolerance of thermal stress, compared with the Vf strain; Vf germination rates at 24 h were reduced by 7.1% (*p <* 0.05) (Figure 6D). Hence, it appears that CchPV1 infection has an important effect on host conidial tolerance of both UV irradiation and thermal stress.

## 4. Discussion

Mycoviruses isolated from *C. chanhua* have not been investigated extensively. To date, only two viruses, one *Alternavirus* CcAV1 and one *Victorivirus* CcV1, have been described in *C. chanhua* [29]. In this study, we report a third novel dsRNA virus infecting *C. chanhua*. To our knowledge, it is the first report of a partitivirus from this medicinal fungus.

In *C. chanhua* isolate RCEF5997, CchPV1 had three dsRNA segments detected by electrophoresis, which revealed the presence of extrachromosomal genetic elements (EGEs). However, the CchPV1 in isolate RCEF5833 lost dsRNA 3 via horizontal transmission from isolate RCEF5997. Furthermore, only two dsRNA fragments are reported in all *Gammapartitiviruses*. Therefore, we speculate that dsRNA 3 is a defective dsRNA from dsRNA 1 or dsRNA 2.

When *C. chanhua* isolates RCEF5997 and RCEF5833 were co-cultivated, a barrage line could be clearly observed between these two strains. Fortunately, we obtained one isolate RCEF5833 with CchPV1. Thus, the mycovirus was successfully transferred to recipient strain RCEF5833. Similarly, co-cultivation resulted in within-species transmission of Cryphonectria naterciae fusagravirus 1 (CnFGV1) to virus-free strains of *C. naterciae*, even if a vegetative incompatibility reaction could be clearly observed between some isolates [30]. Thus, some mycoviruses are able to infect new hosts beyond incompatibility.

We tried to further evaluate effect of CchPV1 on the virulence of host. However, due to the difficulty in rearing cicadae in the laboratory, we used the hind wings of *Platylomia* spp. to conduct penetration assays instead of a bioassay. The results showed that the relative penetration rate of Vi was slightly higher than that of Vf (*p* < 0.05) (Appendix A), suggesting that CchPV1 can improve the ability of *C. chanhua* to infect its host cicada by enhancing penetration. This was similar to the effect of BbPmV4-2 on host *B. bassiana*, whereby mycovirus infection significantly increased fungal virulence against the greater *Galleria mellonella* by enhancing hydrophobicity, adhesion, and penetration of Vi (Shi et al., unpublished)

In summary, some partitiviruses have been identified that modulate the virulence of phytopathogenic fungi [10], but little is known about their roles in the entomogenous fungi [20]. Here, we characterized a novel *Gammapartitivirus* from entomogenous and medicinal fungus *C. chanhua* RCEF5997, which contributes useful information toward a better understanding of mycovirus diversity. Furthermore, this study revealed that the mycovirus causes changes in fungal development and multi-stress tolerance to *C. chanhua*. Importantly, CchPV1 infection increases the conidiation and formation of fruiting bodies of the host. Our work illustrates a novel strategy for producing high-quality fruiting bodies of *C. chanhua*.

## Figures and Tables

**Figure 1 jof-08-01309-f001:**
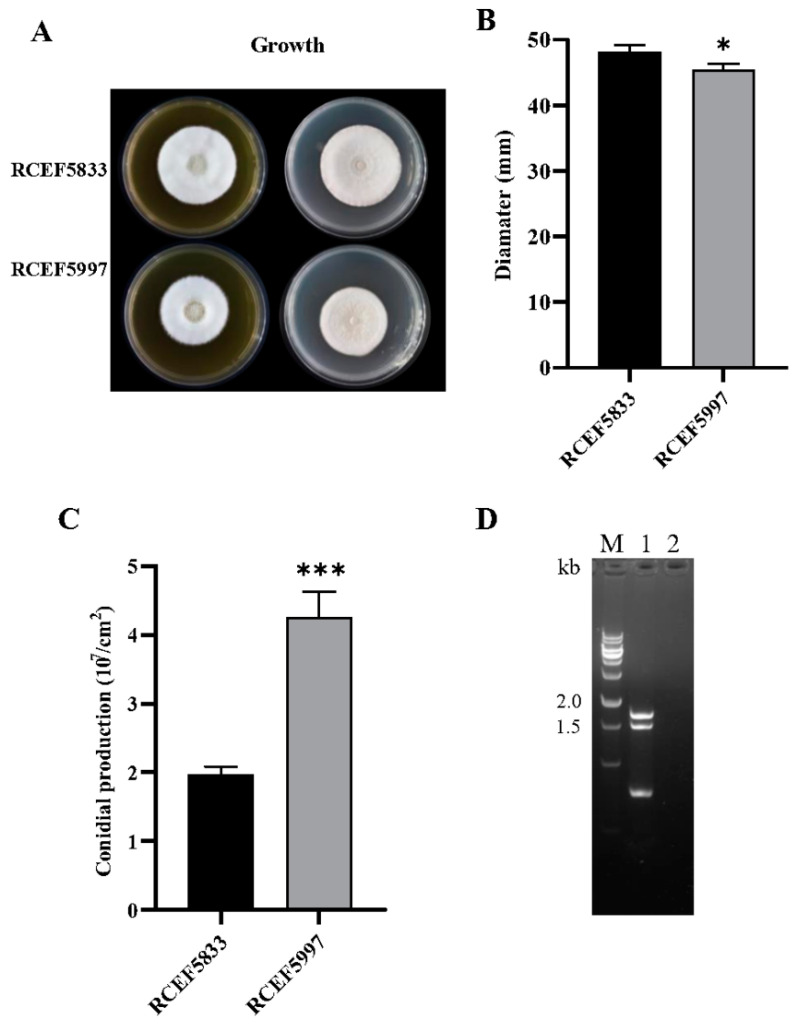
Fungal development and virus infection of *Cordyceps chanhua* strains. (**A**) Colony morphology of *C. chanhua* strains RCEF5997 and RCEF5833 on PDA medium. (**B**) Diameters of strains grown on SDAY for 10 days. (**C**) Sporulation capacity of strains RCEF5997 and RCEF5833 growing on PDA medium after 10 days. (**D**) Electrophoresis of purified dsRNAs extracted from *C. chanhua* strains RCEF5997 and RCEF5833 in a 1.5% agarose gel. M, DNA marker; lane 1, dsRNA segments of RCEF5997; lane 2, RCEF5833 did not possess any dsRNA. * *p* < 0.05, *** *p* < 0.001.

**Figure 2 jof-08-01309-f002:**
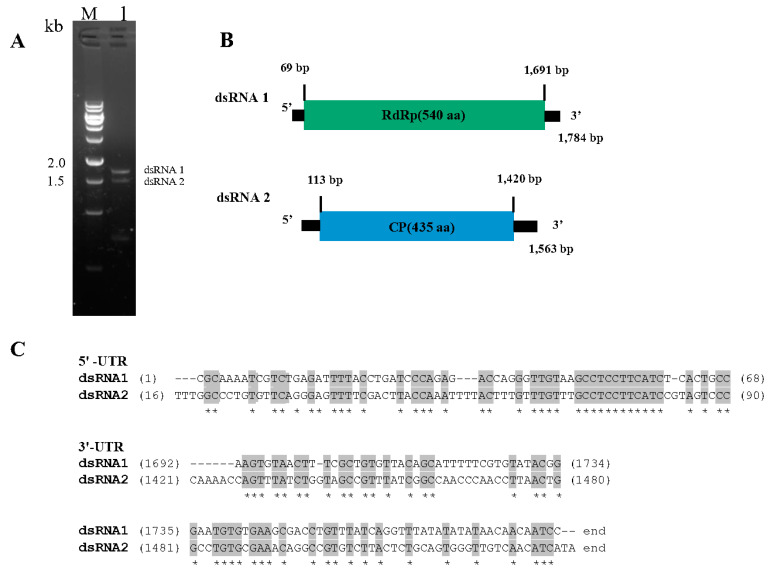
Characterization of CchPV1. (**A**) Electrophoresis of purified dsRNAs extracted from *Cordyceps chanhua* RCEF5997 in a 1.5% agarose gel. M, DNA marker; lane 1, dsRNA segments of RCEF5997. (**B**) Schematic representation of CchPV1. (**C**) Sequence alignment information of 5’-UTR and 3’-UTR of CchPV1 (ClustalX was used for sequence alignment). An asterisk indicates a position with a single, fully conserved residue. Colons and periods indicate conservation between groups with strongly similar properties (scoring > 0.5) and weakly similar properties (scoring ≤ 0.5), respectively, in the Gonnet PAM 250 matrix.

**Figure 3 jof-08-01309-f003:**
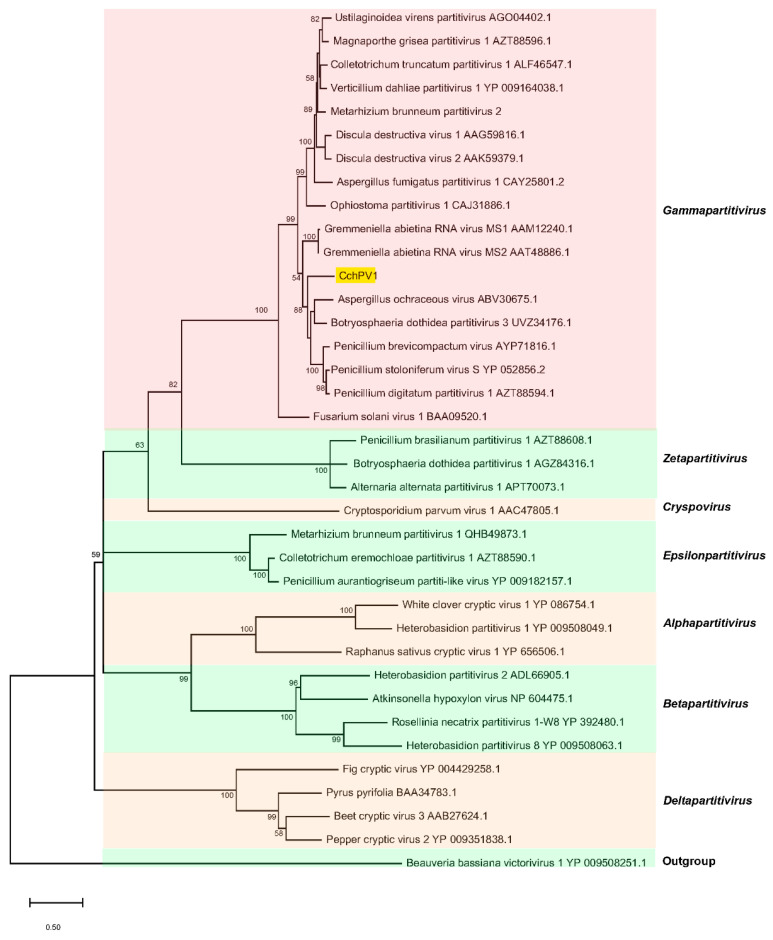
Phylogenetic analysis of RdRp of CchPV1 and the other members of the family *Partitiviridae*, including members of the genera *Alphapartitivirus*, *Betapartitivirus*, *Gammapartitivirus*, *Deltapartitivirus*, and *Cryspovirus*, as well as the proposed genera “*Epsilonpartitivirus*” and “*Zetapartitivirus*”, constructed using the maximum likelihood method (ML), with the LG + G + I + F amino-acid substitution model. The scale represents 0.5 amino-acid substitutions at each site, and the numbers on the nodes indicate bootstrap support of more than 50% (1000 repeats).

**Figure 4 jof-08-01309-f004:**
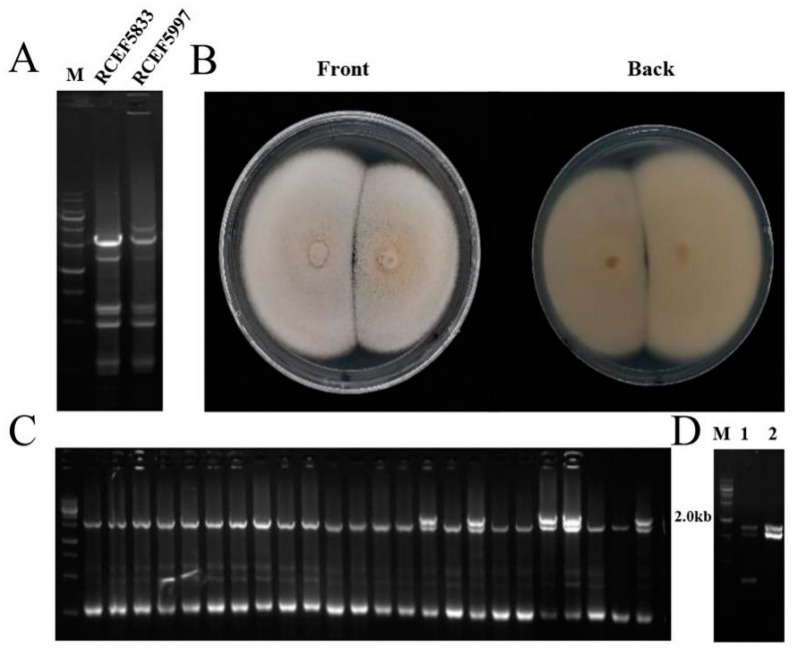
Transmission of CchPV1 virus. (**A**) Strains RCEF5833 and RCEF5997 were genetically distinguished by ISSR markers. (**B**) Confrontation culture of RCEF5997 and RCEF5833, in which the large colony was RCEF5833 and the small colony was RCEF5997. (**C**) ISSR analysis of single-spore isolates. (**D**) dsRNA verification of single-spore isolates, M, DNA marker; lane 1, dsRNA segments of RCEF5997; lane 2, dsRNA segments of RCEF5833 (Vi).

**Figure 5 jof-08-01309-f005:**
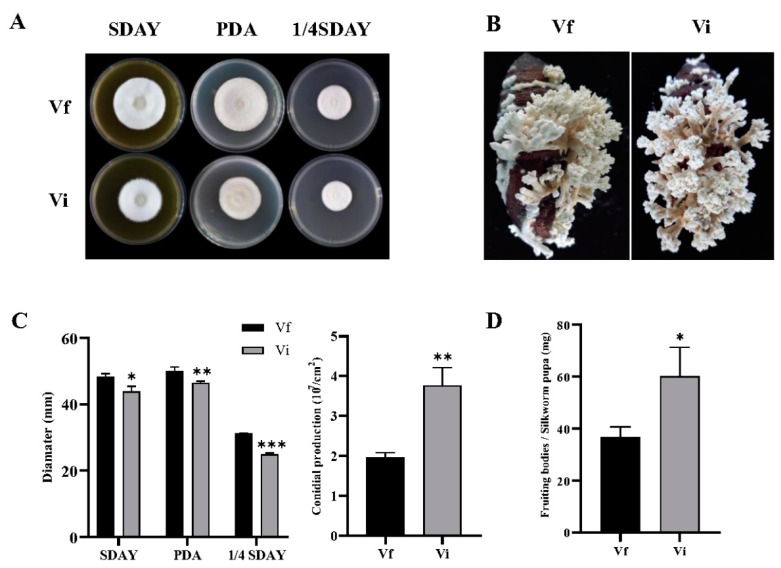
Fungal development of Vf and Vi strains. (**A**) Colony morphology of strains Vf and Vi growing on different media. (**B**) The growth of asexual fruiting on Chinese tussah silkworm pupae. (**C**) Left: diameters of strains grown for 10 days on PDA, SDAY, and 1/4 SDAY, respectively; Right: sporulation of the strains after 10 days of growth. (**D**) The production of the fruiting bodies of 1 g of pupae produced by Vi and Vf strains. * *p* < 0.05, ** *p* < 0.01, *** *p* < 0.001.

**Figure 6 jof-08-01309-f006:**
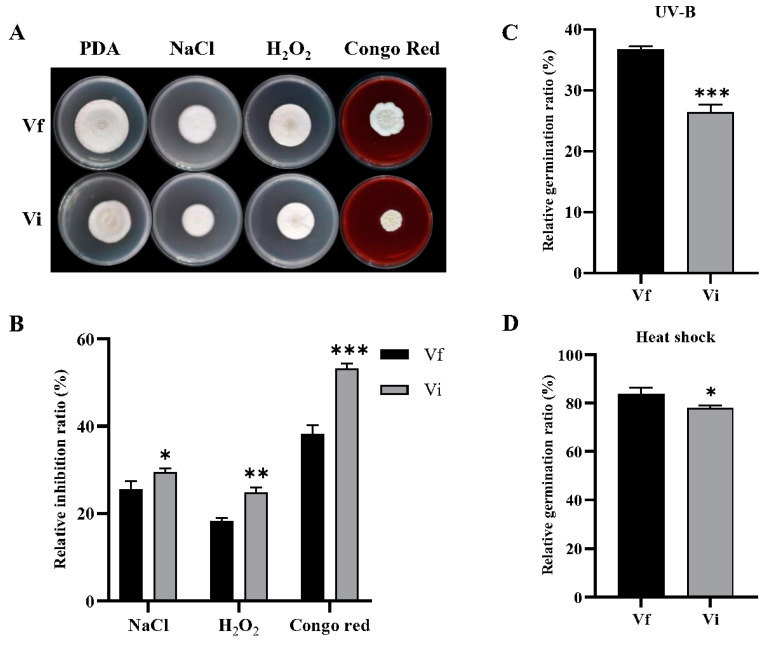
Resistance of strains to stress. (**A**) Colony morphology of Vf and Vi in PDA containing different chemicals. (**B**) The relative inhibition rate of the strains cultured on the medium containing NaCl, H_2_O_2_, and Congo Red after 10 days. (**C**) Relative germination rate of strains after 24 h UV-B irradiation. (**D**) The relative germination rate of strains at 40 °C heat shock after 24 h. * *p* < 0.05, ** *p* < 0.01, *** *p* < 0.001.

## Data Availability

Not applicable.

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
