# Peer review of "A Novel Gammapartitivirus That Causes Changes in Fungal Development and Multi-Stress Tolerance to Important Medicinal Fungus Cordyceps chanhua"

_jof, 2022, doi:10.3390/jof8121309_

Round 1

Reviewer 1 Report

The authors sequenced a Gammapartitivirus from Cordyceps chanhua, conducted phylogenetic analysis. The transferred the virus to a virus-free strain, and compared the isogenic strains for growth and conidiniation. The study is relatively well designed and the presentation was reasonably clear.  

A few comments for improvement:

1. What about virus particle? Since partivirused are capsided, it would be nice to characterize the virus particle.

2. dsRNA3 was not sequenced. The authors could traditional sequencing technology to obtain sequence of this segment. dsRNA seems not transferred to the virus-free isolate (Fig 4D).

3. The ISSR data is not clear to me. Based on which amplified fragment did they determine 61 out of 70 single spore isolates were from isolate 5833? (Fig 4C).

Author Response

Comments and Suggestions for Authors: The authors sequenced a Gammapartitivirus from Cordyceps chanhua, conducted phylogenetic analysis. The transferred the virus to a virus-free strain, and compared the isogenic strains for growth and conidiniation. The study is relatively well designed and the presentation was reasonably clear.

A few comments for improvement:

1.What about virus particle? Since partivirused are capsided, it would be nice to characterize the virus particle.

Response: The descriptions about virus particle have been added. Lines 52-58.

2.dsRNA3 was not sequenced. The authors could traditional sequencing technology to obtain sequence of this segment. dsRNA seems not transferred to the virus-free isolate (Fig 4D).

Response: The dsRNA 3 fragment was not successfully transmitted to the recipient strain.

3.The ISSR data is not clear to me. Based on which amplified fragment did they determine 61 out of 70 single spore isolates were from isolate 5833? (Fig 4C).

Response: A total of 70 single-spore isolates were obtaine from the margin of co-cultivated colony and subjected to ISSR analysis, the results showed that 61 single-spore isolates were identified as RCEF5833 (primer P6 in the supplementary material). However, some ISSR analysis results presented in Fig 4C. And all ISSR data is in Supplementary Table S1B.

Reviewer 2 Report

The manuscript brings new information about the sequence of the novel partitivirus and its effects on its host. I recommend this MS for publication after thorough correction of all errors and revision by a native English speaker.

Please keep in mind that the acronym CcPV1 has been used previously (2010) in 10.1016/j.vetmic.2009.02.002ff. ffhal-00490542 for a papillomavirus!!! It will probably be necessary to change the abbreviation of the gammapartitivirus throughout MS!

Please, follow the Journal´s instructioin for authors: the references should be in brackets [1], not index 1!

Typos:

line 15 - a novel mycovirus

line 48 - entomopathogenic fungi did not represent specific taxon among fungi. Reformulate the sentence.

line 65 - delete "called cryptic viruses"

line 72 - typos "confirmed"

line 76 - "...we found that in two two C. chanhua isolates, RCEF5833 and RCEF 5997..."

line 82 - ...and analysed its...

line 87 - The C. chanhua virus donor strain...

line 88 - ...were isolated from cicadas (wchich ones?) in Anhui Province...

line 97 - the dsRNA were treated...

line 106 - not correct link (blast.ncbi.nlm.nih.gov/Blast.cgi)

line 108 - rapid amplification

line 109 - reformulate: ...introduced unto E. coli TOP10 for sequencing at least three times

line 130 - explain SDAY

line  135 - more probably 6 cm diameter is correct?

line 146 - ...bodies formation assay, 1 mL...

line 147 - reformulate the sentence (missing words there?)

line 162 - (Tokyo, Japan)

line 171 - ...were diluted to ??? with???

line 173 - how the experiment was performed in details? Why the plates were used?

line 183 - C. chanhua isolates...

line 186 - ...2-fold (missing word here?) conidia...

line 189 - ...while no dsRNA was detected...

line 191 - metagenomic sequencing data?? Really? You wrote that the dsRNA was sequenced (line 103)

line 194 - The mycovirus from isolate RCEF5997...

lines 200, 201, 202 - RCEF5833

line 207 - Each segment has a single ORF.

line 208 - a putative 540 aa long ...

line 222 - delete part with : and . explanation as they are not used in Figure 2

line 230 - Moreover, a conserved...

Figure 3 - Add an information what sequence (CP or RdRp) was used for calculation of this tree.

line 326 - ...two viruses...

line 327 - ... have been described... 

line 327 - delete (Zhu et al, 2002)

line 328 - is the first report...

line 350 - according to Instructions, unpublished data shoud be citted also!

Discussion -I miss the comparison of the genomes of CcPV1 with related gammapartitiviruses. I have some doubts about the completeness of the 5' end of the RNA2 segment due to the absence of the very conservative motif CGCAAA, which is present in the RNA1 segment and in almost all gammapartitiviruses. See publications 10.3390/v13112254. or 10.3390/v14020331 for comparison.

References - Follow Instruction rules for authors. Journals names should be in abbreviated form

line 387 - Gilbert

line  389 - Nibert 

Author Response

Comments and Suggestions for Authors: The manuscript brings new information about the sequence of the novel partitivirus and its effects on its host. I recommend this MS for publication after thorough correction of all errors and revision by a native English speaker.

Please keep in mind that the acronym CcPV1 has been used previously (2010) in 10.1016/j.vetmic.2009.02.002ff. ffhal-00490542 for a papillomavirus!!! It will probably be necessary to change the abbreviation of the gammapartitivirus throughout MS!

Response: Thanks for your suggestions. We have corrected in the revised manuscript using CchPV1 as the abbreviation of the gammapartitivirus.

Please, follow the Journal´s instructioin for authors: the references should be in brackets [1], not index1!

Response: Changed as suggested. We have corrected thoroughly.

Typos:

line 15 - a novel mycovirus

Response: Changed as suggested. Line 15.

line 48 - entomopathogenic fungi did not represent specific taxon among fungi. Reformulate the sentence.

Response: Changed as suggested. Line 48.

line 65 - delete "called cryptic viruses"

Response: Changed as suggested. Line 67.

line 72 - typos "confirmed"

Response: Changed as suggested. Line 75.

line 76 - "...we found that in two C. chanhua isolates, RCEF5833 and RCEF 5997..."

Response: Changed as suggested. Line 79.

line 82 - ...and analysed its...

Response: Changed as suggested. Line 85.

line 87 – The C. chanhua virus donor strain...

Response: Changed as suggested. Line 90.

line 88 - ...were isolated from cicadas (which ones?) in Anhui Province...

Response: Changed as suggested. Line 91.

line 97 - the dsRNA were treated...

Response: Changed as suggested. Line 100.

line 106 - not correct link (blast.ncbi.nlm.nih.gov/Blast.cgi)

Response: Changed as suggested. Line 110.

line 108 - rapid amplification

Response: Changed as suggested. Line 112.

line 109 - reformulate: ...introduced unto E. coli TOP10 for sequencing at least three times.

Response: Changed as suggested. Lines 113-114.

line 130 - explain SDAY

Response: Changed as suggested. Lines 134-135.

line 135 - more probably 6 cm diameter is correct?

Response: Changed as suggested. Line 140.

line 146 - ...bodies formation assay, 1 mL...

Response: Changed as suggested. Line 151.

line 147 - reformulate the sentence (missing words there?)

Response: Changed as suggested. Line 153.

line 162 - (Tokyo, Japan)

Response: Changed as suggested. Line 168.

line 171 - ...were diluted to ??? with???

Response: Changed as suggested. Line 177.

line 173 - how the experiment was performed in details? Why the plates were used?

Response: C. chanhua can be grown on both SDAY plates, so the SDAY plates were used. Lines 176-182.

line 183 -C. chanhua isolates...

Response: Changed as suggested. Line 192.

line 186 - ...2-fold (missing word here?) conidia...

Response: Changed as suggested. Line 194.

line 189 - ...while no dsRNA was detected...

Response: Changed as suggested. Line 197.

line 191 - metagenomic sequencing data?? Really? You wrote that the dsRNA was sequenced (line 103)

Response: Changed as suggested. Line 198. RNA-sequencing was used for dsRNAs sequencing.

line 194 - The mycovirus from isolate RCEF5997...

Response: Changed as suggested. Line 202.

lines 200, 201, 202 - RCEF5833

Response: Changed as suggested.

line 207 - Each segment has a single ORF.

Response: Changed as suggested. Line 215.

line 208 - a putative 540 aa long ...

Response: Changed as suggested. Lines 217.

line 222 - delete part with : and . explanation as they are not used in Figure 2

Response: Changed as suggested.

line 230 - Moreover, a conserved...

Response: Changed as suggested. Lines 238.

Figure 3 - Add an information what sequence (CP or RdRp) was used for calculation of this tree.

Response: Changed as suggested. Lines 256. The sequence used for the tree computation is RdRp.

line 326 - ...two viruses...

Response: Changed as suggested. Lines 334.

line 327 - ... have been described... 

Response: Changed as suggested. Lines 335.

line 327 - delete (Zhu et al, 2002)

Response: Changed as suggested.

line 328 - is the first report...

Response: Changed as suggested. Lines 336.

line 350 - according to Instructions, unpublished data shoud be citted also!

Response: Changed as suggested.

Discussion -I miss the comparison of the genomes of CcPV1 with related gammapartitiviruses. I have some doubts about the completeness of the 5' end of the RNA2 segment due to the absence of the very conservative motif CGCAAA, which is present in the RNA1 segment and in almost all gammapartitiviruses. See publications 10.3390/v13112254. or 10.3390/v14020331 for comparison.

Response: The 5′ UTRs of dsRNA 1 and dsRNA 2 have the conservative motif CGCAAA (Supplement data 1).

References - Follow Instruction rules for authors. Journals names should be in abbreviated form

line 387 - Gilbert

Response: Changed as suggested.

Line 389 - Nibert

Response: Changed as suggested.